# Quasi-Seniors’ Perception, Response, and Planning from the Perspective of Successful Aging

**DOI:** 10.3390/healthcare12070766

**Published:** 2024-03-31

**Authors:** Ming-Shien Wen, Miao-Hsien Chuang, Jinkwan Lin

**Affiliations:** 1Division of Cardiology, Chang Gung Memorial Hospital, Linkou Branch, Taoyuan 105406, Taiwan; wenms123@cgmh.org.tw; 2School of Medicine, Chang Gung University, Taoyuan 333323, Taiwan; 3Department of Visual Communication Design, Ming Chi University of Technology, New Taipei 243303, Taiwan; 4College of Management and Design, Ming Chi University of Technology, New Taipei 243303, Taiwan; jinkwan@mail.mcut.edu.tw

**Keywords:** successful aging, quasi-seniors, adaptation theory of aging, aging-integrated

## Abstract

With the coming of a rapidly aging society, individuals born in the baby boom era after World War II are now facing the challenges of aging. From late middle age to successful aging, what are the perceptions and responses of these quasi-seniors? With this in mind, referring to Phelan’s successful aging scale, the researchers developed the 4P Strategies (Physical, Psychological, Prospect, and Place and Relationships) tailored for quasi-seniors. Based on grounded theory, the results of 12 sessions of focused interviews (involving a total of 93 interviewees between the ages of 55 and 75; 41 males and 52 females; 48 not retired and 45 retired) were matched with the 4P Strategies. The results were the following: (1) regarding the Physical factor, the interviewees were shocked by their physical decline, and they had begun to devise strategies for health preservation and exercise; (2) regarding the Psychological factor, in order to mentally adapt, the interviewees agreed that moderate stress relief was absolutely necessary; (3) regarding the Prospect factor: the interviewees felt that one should make financial plans early, contemplate the value of life, and more actively learn and realize one’s dreams; and (4) regarding the Place and Relationships factor, the interviewees aimed to rebuild their close relationships with their spouses, family members, and old friends and had polarized views regarding where to live in their old age. On the whole, the most discussed issue among the interviewees was where to live in their old age. Many had their own views and plans and did not stick to traditional views; however, they took the opinions of their significant others into account. During the interviews, interviewees wished to understand the responses of their peers to serve as a reference for their own decisions, and they realized that successful aging also required learning. This study aimed to encourage quasi-seniors about to enter their old age and help them to learn how to positively respond to aging as well as work towards a happy life with successful aging. This study could fill in gaps in research involving individuals in this age group and provide a reference for relevant policies.

## 1. Introduction

The World Health Organization (WHO) has stated that individuals over the age of 65 are classified as seniors and that population aging is an indisputable fact, which presents a major challenge for every country around the world. The UN has even declared that the period 2021–2030 is the Decade of Healthy Ageing and has requested that the WHO take the lead in collective actions to reduce health inequality and improve the family and community life of seniors.

Old age can be divided into three stages; seniors between the ages of 65 and 74 are “young-old”, those between the ages of 75 and 84 are “old-old”, and those over the age of 85 are “older-old” [1]. The baby boomers, born between 1946 and 1964 after World War II (WWII), are one group of people that cannot be ignored, as they are already seniors or are about to become seniors. Statistics show that almost 76 million people were born in the US during that period, and although researchers disagree about which years the post-WWII baby boom in Taiwan spans, the population born around this period is still substantial, which includes late middle-aged and young-old people. Whether it is through their educational background, attitudes, preferences, or behaviors, these individuals exert a profound impact on the economy, the workplace, and the entire social system. In general, they have better educational backgrounds and enjoy better economic environments and living standards than their parents. Physically, they have not degenerated substantially; however, in response to the significant increase in their media literacy as a result of computers, communications, and consumer electronics, they prefer holistic healthcare; that is, in addition to physical health, they also pay attention to psychological, social, and spiritual aspects [2,3].

The most widely discussed topics in geriatrics are healthy aging, successful aging, productive aging, and active aging [4,5,6]. Among these, successful aging garners the most widespread acclaim, with the definition provided by Rowe and Kahn [7,8] being the most frequently referenced. Indeed, the term “aging” has long been equated with negative stereotypes such as illness, loneliness, disability, and care arrangements. Rowe and Kahn believed that successful aging has the following three key behaviors or characteristic abilities: (1) physiologically, avoiding disease and disability; (2) psychologically, maintaining high cognitive and physical function; and (3) socially, engaging actively in social life, which includes two primary elements, namely social relations and productive activities. The latter has three predicting factors, namely functional capacity, education, and self-efficacy. The intersection of behaviors in the physiological, psychological, and social aspects above is successful aging. Furthermore, there are also three theories on coping with aging, as follows: disengagement theory, activity theory, and continuity theory. The activity theory holds that seniors who can maintain social activities can better obtain a positive sense of self, social integration, and life satisfaction. Baltes and Baltes [9] asserted that, through the process of selection, optimization, and compensation, seniors maximize their gains with minimal losses in response to the impacts of aging.

In addition to the discourse by the aforementioned scholars, Phelan et al. [10], based on an elderly centered perspective, found that the definition of successful aging was of utmost importance to research in the field, from their viewpoint. In Phelan’s survey [11] of Japanese Americans and Caucasian individuals, she found that approximately six out of ten interviewees experienced changes in their views over the past twenty years. The Japanese Americans rated 13 items as a part of successful aging; in addition to the same 13 items, the Caucasians included another item—learning new things. Phelan concluded that, for seniors, the definition of successful aging is multidimensional and includes four major categories: physical health, functional health, psychological health, and social health.

Regarding the life course, Erikson [12] delineated human life into eight major stages. At 65 years old and onward is the “Integrity vs. Despair” stage. The development tasks and crises of this stage are self-pride (integrity) and despair. During this stage, individuals often look back on their lives, including what they have accomplished, what kind of person they are, and whether their lives have been meaningful. They are satisfied with their efforts and achievements, which leads to a sense of balance, wholeness, and integrity. At the same time, however, they face aging, disease, and the reality of the time that they have left, which leads to sorrow, despair, and regret. Nonetheless, Riley and Riley [1] proposed the concept of age integration, breaking the norms regarding the role that age plays in different stages of life. In the past, people tended to act in accordance with social norms. For instance, they received education during their adolescence, bore work and family responsibilities in their middle age, and enjoyed a leisurely retirement in their old age for granted. In the structure of today’s rapidly aging societies, this age-differentiated approach makes it difficult for seniors to adapt to their roles properly, thereby creating structural lag.

The increase in the average life expectancy of human beings, as assessed by Yang [13], has lengthened the period deemed middle age. She appealed for the Third Act, which was proposed by Dr. Edward Kelly. The Third Act refers to the third act of life, and “act” can also serve as a verb, implying that nothing will be gained if there are no actions during this part of life. The Third Act divides life into the following three phases: (1) dependence, the growth period in which individuals absorb and depend on the nutrition and nourishment provided by those around them in order to grow; (2) independence, the period during which individuals form their own families, build their own careers, and begin to create more possibilities on their own; and (3) inter-independence, the period during which individuals may require some assistance but still have the ability to give back to society, forming symbiotic and co-prosperous relationships with others and the environment.

An overview of existing geriatric research reveals a scarcity of studies on quasi-seniors born during the baby boomer period, despite the variation in seniors’ perspectives across different stages. In contrast to Lu’s approach [14], who used subjective/objective and end-state/process dimensions for categorizing theories on geriatric welfare and positive aging, this research refers to Phelan’s four-dimensional framework [11]. It converts the passive/static adaptation of the functional factor into the active/prospective planning of quasi-seniors and examines their perceptions, responses, and living plans regarding physical and psychological signs of early aging. This study utilizes focus interviews to shed light on the physical and psychological evaluations and preparations made by individuals aged 55 to 75, born in the post-WWII baby boom era, as they transition into old age. It aims to explore how these quasi-seniors overcome age-related challenges, actively develop coping strategies, and endeavor towards successful aging.

## 2. Methodologies

### 2.1. Selection of Methodologies

Qualitative studies aim to understand the experiences of the subject as well as how subjects themselves interpret these experiences and how they construct the social world in which they live [15]. Narration is defined as discourse aimed at certain audiences and the sequential presentation of related events using meaningful methods in order to provide an understanding of the world and/or the experiences of people [16]. When contemplating how to reach, interpret, and analyze life stories and various possibilities of other narrative materials, Lieblich et al. [17] discovered two independent orientations and developed four quadrants, the holistic vs. categorical orientation and the content vs. form orientation. Here, the categorical/content orientations were adopted, in which studied topic categories have been clearly defined. The separated paragraphs in this paper were extracted, categorized, and then reassembled into these categories/clusters. The contents of the focused interviews were placed into Phelan’s four categories of successful aging and then converted into four categories of quasi-senior responses to aging.

This study was primarily a qualitative investigation. Our methodologies included focused interviews, grounded theory analysis, and t-tests, which are briefly explained below:Focused interviews: A focus group is a form of qualitative investigation method that involves six to twelve participants engaging in free and interactive discussions on certain topics to collect deeper and more authentic views and opinions. The objective is to observe a large amount of verbal interaction and dialogue within a short period of time [18]. In this study, focused interviews were conducted with individuals between the ages of 55 and 75. Each session involved seven to eleven interviewees, and the discussions were recorded and transcribed for grounded theory analysis.Grounded theory analysis: This included data processing, categorization and conceptualization, context establishment, and interpretation.(1)Data processing: This included numbering the interviewees, the pages of the transcripts, and the conversations between the interviewer and the interviewees.(2)Open coding: This included transcribing the interviews, reviewing and analyzing the transcripts, and performing open coding.(3)Categorization, comparison, and conceptualization: After all of the transcripts were coded, the compiled documents were categorized, examined, compared, and conceptualized. Those with similar concepts were categorized to develop a core category. The verbalizations in the self-reports of the interviewees were listed to corroborate and explain the concepts or framework created by the researcher.(4)Establishment of context among concepts: The researcher placed the conditions, contexts, actions, or interaction strategies of the analyzed phenomena with the results and identified the axial coding through the coding paradigms among the categories to facilitate a better grasp of the context among the various concepts.(5)Interpretation: Through theoretical sensitivity or interpretation, the collected data were explained in greater depth using established theories. We transcribed the contents of the focused interviews, selected 918 key statements, and identified the primary and secondary factors.

### 2.2. Reliability and Validity

Reliability: To test reliability, researchers distributed relevant information to the participants and gave them the chance to comment, add information, change their minds, and provide their own interpretations.

Validity: Internal validity and external validity increase overall data validity. The interviewees participated in discussions, and the entire process was recorded in case the notes taken were inadequate. The data were re-confirmed and verified, and proofreading procedures were implemented carefully.

Triangulation: This refers to the use of different methods, data, observers, and theories during studies as well as applying and combining several types of methods, theories, researchers, or data sources to overcome biases that may result from a single method, researcher, theory, or data source. This study combined multiple theories associated with old age, data from the 12 sessions of focused interviews, and multiple raters to give consideration to both reliability and validity.

### 2.3. Study Participants and Procedures

Study participants: This study recruited participants from locations such as Chang Gung Health Village and senior community colleges in Taiwan. Prospective participants were required to be voluntary, able to articulate clearly verbally, free from serious health issues, and capable of independently performing daily activities. The study participants were between 55 and 75 years of age. They included 41 men and 52 women. In terms of retirement status, 45 were retired, and 48 were not retired. Among the participants, 37 lived in retirement communities, and 56 lived in their current place of residence.Data collection and organization: The data from the 93 participants in the 12 sessions of focused interviews were transcribed, and key points were extracted. For the conceptualization process, this study referred to the definition provided by Phelan et al. [11] on successful aging, and open coding was performed to identify the secondary factors. Following a process of selection, classification, comparison, and interpretation [19], meaningful statements were extracted and coded. The coding principles for each interviewee number were as follows. The first set of two digits was the number of the interviewee. In the second set of digits, the first two digits indicated the number of the focused interview session, and the last two digits represented the number of the interviewee within this session. The digit in the third set indicated the gender of the interviewee (1: male; 2: female). The digit in the fourth set represented the retirement status of the interviewee (1: not retired; 2: retired), and the digit in the fifth set was the interviewee’s place of residence (1: home or current residence; 2: retirement community). For example, 17-0302-2-1-1-0169 means that the interviewee data and text source were from Interviewee No. 17, who was No. 2 in Session 03 of the focused interviews; the interviewee was female, was not retired, and lived in her current residence. The extracted text began on Line 0169 of the transcript of this session.Reliability test of coded data: Inter-rater reliability was adopted for reliability analysis. The researcher and two university teachers served as the raters. After coding all of the data, the researcher made a coding list. A computer then randomly selected 50% of the data, and the other two raters performed reliability tests independently, according to the coding list, to ensure the consistency of the coding results. The coding results from the researcher and the two other raters were tested using the percent agreement formula. The results were 0.96 in the Physical factor, 0.94 in the Psychological factor, 0.84 in the Prospect factor, and 0.81 in the Place and Relationships factor, thereby indicating that the coding results in this study were reliable.

## 3. Results

The four major categories that Phelan established for successful aging were modified in this study to Physical, Psychological, Prospect, and Place and Relationships, based on the perceptions and responses of quasi-seniors. Using the first letter of these categories, they are referred to as the 4P Strategies of Quasi-seniors and explained accordingly below:

### 3.1. Responses of Interviewees to Physical and Psychological Aspects of Aging

#### 3.1.1. Physical Items

The perceptions and responses regarding the body can be divided into (1) physical changes, (2) health preservation, and (3) health and exercise. The representative statements made by the interviewees were as follows.

Physical Changes (P1)

Regardless of how well they performed in sports or at work in the past, the interviewees were generally aware that their bodies were deteriorating, to the point of affecting their confidence. Some representative statements were as follows:
*I’m almost 60. I don**’t feel old, but my stamina tells me I’m old*.*(03-0103-1-1-1-0010)*
*Even drinking water makes me fat*.*(01-0101-1-1-1-0165)*
*My eyesight is failing. My concentration is failing. I forget things I’ve learned; it’s a bit embarrassing*.*(36-1103-1-2-1-0046)*
*I like to hike and exercise. I ran 68 marathons, but I stopped due to fasciitis*.*(45-1203-1-1-1-0211)*

2.Health Preservation (P2)

Eating mild and low-calorie diets was the general consensus. Some representative statements were as follows:
*Less oil, less salt, less sugar, less processed foods, best boiled, nuts*.*(52-1302-1-2-1-0280)*
*Go on a diet, and eat a light and non-greasy diet*.*(09-0201-1-1-1-0283)*
*Only light diets are healthy. But the question is: can you stick to them?**(42-1109-2-1-1-0604)*
*I’ve cared for my diabetic mother and aunt for eight years. I have experience with health preservation, and it’s true that you have to be mindful of what you eat*.*(46-1204-1-1-1-0405)*
*My wife controls what I eat. She buys supplements, and I have to take them*.*(07-0107-1-1-1-0225)*

3.Health and Exercise (P3)

The interviewees felt that nothing was more important than health and exercise. Some even felt that it was worth sacrificing some of their achievements.
*You can ask anyone living in a retirement community, and they’ll go on about how rich they are and so on, but I tell you: health is the most important*.*(63-0408-2-2-2-672)*
*You have to exercise if you want to preserve your health. A bit of exercise and sweating are probably better than taking any supplements*.*(14-0207-1-1-1-0362)*
*Spend time on your health after you’re 30, and you’ll see a big difference between you and your classmates in your 60s and 70s. My diet, exercise, daily habits… if I could do it all over again, I’d spend more time on these, even if I had to sacrifice some of my achievements*.*(90-0804-1-2-2-0660)*

#### 3.1.2. Psychological Items

The perceptions and responses regarding the body can be divided into (1) psychological perception, (2) easy stress handling, and (3) proactiveness and positivity. Some representative statements made by the interviewees were as follows:Psychological Perception (M1)

Some interviewees exclaimed outright that they did not think of the day they would get old. Others realized that they were aging due to a friend or family member passing away or due to information from the media.
*How did I get old, too? I had never thought about that*.*(21-0306-2-1-1-0229)*
*I still can accept that I’m getting old, my hair is going gray, and my teeth are becoming loose*.*(10-0202-2-1-1-0091)*
*I said goodbye to five friends last year (who passed away); they were all in their early 50s*.*(28-1006-1-1-1-0070)*
*After you retire, you grow apart from your coworkers and classmates, and then there’s nothing at the end. You’ll only have a few of your closest friends*.*(39-1106-1-2-1-0540)*

2.Easy Stress Handling (M2)

The interviewees felt that going with the flow and relieving stress moderately were the most important. Some representative statements were as follows:
*Having the right mindset and attitude is more important than taking supplements*.*(12-0204-1-1-1-0254)*
*I have no plans and live in the now. I just need food in my belly. I’m a freelancer right now*.*(44-1202-1-1-1-0255)*
*I treat myself well. I buy myself a small gift on my birthday. I haven**’t learned how to have coffee myself, though*.*(18-0303-2-1-1-0026)*
*Don’t stress! Enjoy the new experiences that you now have first*.*(24-1001-1-1-1-0102)*

3.Proactiveness and Positivity (M3)

The interviewees felt that one should plan early and enjoy continuing to work. There were also references to the diligent work image of older generations. Some representative statements were as follows:
*I think the period between 55 and 75 years old is the most important part of life*.*(29-1007-1-1-1-0023)*
*Plan your ideal old age; don’t wait until you’re retired to think about it*.*(24-1001-1-1-1-0542)*
*If there’s work we can do, we still like to work half the day and then use the rest of the time for leisure. Having fun all 365 days of the year can get boring, too*.*(52-1302-1-2-1-0032)*
*I’ll work until I’m old and can move no longer. My father was the same*.*(29-1007-1-1-1-0062)*

### 3.2. Views of Interviewees Regarding Prospect as well as Place and Relationships of Aging

#### 3.2.1. Prospect Factor

The grasp and plans of the interviewees with regard to the Prospect aspect can be divided into (1) financial plans, (2) learning and dreams, and (3) meaning. Representative statements made by the interviewees are presented below.
Financial Plans (F1)

The interviewees agreed that finances were important and believed that they could only be at ease if they had enough savings; they also shared their ways of estimating retirement savings. A selection of representative statements was as follows:
*It doesn’t matter if you get married or have children; you’re alone in the end, so you have to make plans no matter how young you are. When you are pursuing your career between 40 and 60, don’t forget your health, and save some money so that you can have a good life in your old age*.*(69-0505-2-2-2-0732)*
*I suggest putting your finances and health first; you don’t have to be perfect at work*.*(79-0609-2-2-2-0722)*
*Be good to yourself. Definitely be financially independent, and you’ll have a high social status and naturally gain the respect of others*.*(91-0805-2-2-2-0630)*

2.Learning and Dreams (F2)

Many of the interviewees mentioned their dreams and learning engagements. Some representative statements were as follows:
*Right now, I’m planning to realize my dreams one by one. I want to go to Paris for a six-month course and go to New York to go skydiving, diving, and take pictures*.*(35-1102-2-2-1-0170)*
*If you want to do something, do it now. Don’t wait until you’re old to realize your dreams. Do it now, and do it yourself. Set realistic goals first. Don’t travel the world first. Start by traveling around one place. If your dream is too big, start little by little*.*(66-0502-2-2-2-0135)*
*I wanted to when I heard the urge, but at the time, I didn’t have the courage, the money, or the ability. Then I had the money prepared, but I didn’t have the strength to roam*.*(65-0501-1-2-2-0683)*
*You have to leave the house and take some courses in order to be happy and so your children can be rest assured*.*(75-0605-2-2-2-0624)*

3.Meaning (F3)

The interviewees reflected on the meaning of life and wished to explore their beliefs and live a life of value. Some representative statements were as follows:
*I lived in Australia for 46 years, from being a student to working, getting married, having children, raising my children, and getting divorced. Talking about this makes me feel like a walking robot*.*(67-0503-2-2-2-0656)*
*My brother wasn’t very old, but he died suddenly, and it makes you think about things*.*(32-1010-1-1-1-0187)*
*The thing I’m most afraid of is going through life without having benefited anyone and dying without anyone knowing*.*(68-0504-1-2-2-0398)*
*My ideal life in old age: exploring a faith*.*(15-0208-1-1-1-0643)*
*I try to help the disadvantaged and donate as much as possible*.*(51-1301-2-2-1-0158)*
*Where my value in the future lies is doing something good for kids*.*(21-0306-2-1-1-0241)*

#### 3.2.2. Place and Relationships Factor

The grasp and plans of the interviewees with regard to the Place and Relationships aspect can be divided into (1) friends and family, (2) residence in old age, and (3) significant others. representative statements made by the interviewees are presented below.
Friends and Family (S1)

Many interviewees cared about how they would get along with their spouse after retirement, and they also mentioned their relationships with their siblings, children, and grandchildren. They felt that they needed to rebuild close relationships with friends and family members. A selection of representative statements was as follows:
*Rebuilding close relationships, building close relationships…It’s not just your spouse; there are also your friends and family. I think it’s pretty important. It’s the source of your happiness*.*(05-0105-1-1-1-0499)*
*Well! After retiring, I’m starting over with my husband and whatever because we haven’t been together all day and all night in a very long time*.*(08-0108-2-1-1-0477)*
*I don’t want to become a burden to my children*.*(30-1008-1-1-1-0826)*
*Taking care of your grandchildren may make you feel “useful” in your old age, but instead I want to escape (escaped dream). I don’t want to spend my “seenager” years taking care of my grandchildren*.*(3-0602-2-2-2-0181)*
*Don’t be an annoying old person*.*(82-0703-2-2-2-0899)*
*Just mind your own business*.*(86-0709-2-2-2-0488)*

2.Residence in Old Age (S2)

Younger interviewees wished to live with their family or siblings. However, some felt it was time to and were not against living in a retirement community, believing that retirement communities were safe and organized a lot of activities. Still, they did have to consider their budget. Some representative statements were as follows:
*I’m still going to live in my home in Taipei. It’s bustling whenever I go out, and that makes me happy*.*(02-0102-1-1-1-0601)*
*Ideally, I’m going back to Yilan to buy a house with a yard I can grow vegetables in*.*(45-1203-1-1-1-0222)*
*I’m a single mother. My older sister is not married either. Everyone’s getting old. We siblings want to live in a place to ourselves and take care of each other*.*(42-1109-2-1-1-0486)*
*If I’m mobile, then I want to live by myself. If my health is failing, I’ll tell my kids to send me to a nursing home*.*(16-0301-1-1-1-0218)*
*Hiring a foreign domestic helper is definitely not as good as any professional institution, where I could have friends, take part in activities, and not be a burden to my children. If I really have any health issues, they (the professional institutions) have doctors*.*(03-0103-1-1-1-0584)*
*I’m old. It’s dangerous not having anyone around me, so I’m moving back retirement home again*.*(63-0408-2-2-2-0056)*
*You don’t have to live there forever. You can live in both Taiwan and the US. But live in a good environment for the beginning of the rest of your life*.*(80-0701-1-2-2-0239)*

3.Significant Others (S3)

The interviewees took the opinions of their own elderly parents and their married children into account. Some shared how society views retirement communities and what their own adaptation experiences were. Some representative statements were as follows:
*My dad would not accept living in a retirement home in Tamsui. He was willing to live with me. My mother lives alone. She has her fellow apprentices and doesn’t want to live with me*.*(44-1202-1-1-1-0626)*
*My dad went to check out a retirement home. He felt there was no warmth. It was cold, and there were all old people there*.*(06-0106-2-1-1-0738)*
*My wife says she is willing to live in a retirement community. I haven’t thought about it*.*(15-0208-1-1-1-0637)*
*My younger brother convinced my mom to move into a retirement home. At first, she cried whenever my brother visited. Now, she’s changed her mind and is very thankful. She**’s gradually feeling happier and getting to know a lot of people*.*(62-0407-1-2-2-0481)*
*At the request of her younger brother, the daughter of a military man moved her whole family to Paraguay. Her husband is quite shy, so she handled everything. Her husband likes retirement communities, and her friend came too, so they came back*.*(74-0604-2-2-2-0221)*
*At the time, her in-laws objected to living by themselves. They were going to bring them here first, but they didn’t have the time. It was a rare opportunity, so we moved in*.*(71-0508-2-2-2-0567)*
*Those sent to a nursing home by their children are usually depressed. And if they don’t have any religion or anything to find comfort in, then it’s easier for them to get sick*.*(76-0606-2-2-2-0616)*

## 4. Discussion

### 4.1. Qualitative Research Conclusions and Discussion

Focused interviews were conducted to analyze the four major aspects of which quasi-seniors were aware, with regard to aging and their corresponding factors. During the interviews, many interviewees paid attention to the opinions of others, which served as references from which they could learn. The contents of the interviews were analyzed based on relevant theories. The results and discussions are as follows.

#### 4.1.1. Physical Factor

Physical changes: No matter how much the interviewees paid attention to exercise or could handle social functions in the past, they were generally aware that their bodies were deteriorating, including their memory and stamina. They were surprised that old age was upon them, to the point of affecting their confidence. Regarding the self-awareness of aging, Sato [20] mentioned that seniors perceived their aging from the inside out. Because of failing eyesight, hearing, and memory, they felt themselves degenerating physically and mentally.Health preservation: Beginning to eat light diets was a widespread consensus among the interviewees. The spouses of the male interviewees, in particular, constantly reminded them about this and even controlled their diets. Some were reminded of their mortality by cases in their family, and some doubted their own ability to stick to such diets.Health and exercise: This secondary factor was the most frequently discussed in the Physical factor. The interviewees agreed that nothing was more important than health and exercise. That is, to live, you must move. The retired interviewees even realized that exercise would have been worth sacrificing some of their achievements. Based on the 2018 City and County Aging Development Index questionnaire designed by the Taiwan Active Aging Association, all of the interviewees were asked what they thought the necessary condition for a happy life was, and the most frequent response was also physical health (44.4%).

#### 4.1.2. Psychological Factor

Psychological perception: Some interviewees sighed about aging or were unwilling to accept it. Some perceived that they were aging from friends or family members passing away or from information in the media. Some felt that they were psychologically retired, while others expressed weariness with regard to stress at work, and still others who were retired perceived that their social circle was shrinking. These echo the opinions of other researchers. Kugaya [21] indicated that wishing to remain young and not grow old is a basic biological desire and a natural response supporting the original human desire for self-realization. About 75% of the interviewees felt that they were “unexpectedly young”. Fishman [22] cited the finding of Jim Sherman, indicating that even if they were old, many people still did not believe that old age was upon them. Sato [20] observed that people also perceive their own aging when they see the growth of children and grandchildren, reach retirement age (or transfer), or are treated as a senior.Easy stress handling: This secondary factor was mentioned the most among the psychological factors. The interviewees felt that going with the flow, relieving stress moderately, and relaxing were the best strategies. They expressed that there was no harm in enjoying everyday life, the little joys in life, and even new experiences.Proactiveness and positivity: The interviewees felt that one should plan early, seize one’s golden years, and enjoy continuing to work, as retiring and idling or having fun all day was also boring. There were also references to the diligent work image of older generations, who served as good role models. Krouk [23] also clearly indicated that maintaining creative activities can make people feel proficient, significantly improve their entire health condition, and reduce the level of senior depression and sense of loneliness.

#### 4.1.3. Prospect Factor

Financial plans: The interviewees agreed that finances were important, believing that they should plan early, be financially independent, and have enough savings in order to be at ease and be respected. They also cared about how much funds they needed in order to retire. Yang [13] cited an analysis performed by Jo Ann Jenkins, which indicated that living a meaningful life for 50-year-olds nowadays begins with health, wealth, and self.Learning and dreams: There were lively discussions about this secondary factor, which was also the most discussed item in the Prospect factor. Many of the interviewees mentioned their dreams and the things that they were learning, such as languages, musical instruments, and drawing, which they did not have the opportunity to learn when they were young. There was also hiking, yachting, doing the Grand Tour, among a wide variety of other activities. Some interviewees mentioned that engagement in such activities was also a way of easing the burden on the next generation. Tsai et al. [24] also stated that social involvement can not only help seniors to maintain their actual and perceived health, but it also exerts a positive impact on their mental health and subjective sense of wellbeing. Learning engagements offer knowledge, abilities, and skills. They also provide mental stimulation and re-preparation of physical abilities as well as restore and renew vitality and promote survival abilities in increasingly complex environments.Meaning: The interviewees reflected on the value and meaning of life. Some lamented the mundaneness of the early years of their life, some wished to explore their beliefs, and some had begun contemplating due to the sudden death of a family member. An analysis conducted by Kaufman [25] indicated that seniors are not considered “old” by their friends or family and that they do not consider themselves “old” either, as long as they remain meaningfully active and productive. Takkinen and Ruoppila [26] advocated that the meaning of life is the foundation of a sense of wellbeing in seniors.

#### 4.1.4. Place and Relationships Factor

Friends and family: Many interviewees cared about how they would get along with their spouse after retirement and also mentioned their relationships with their other family members, such as parents, siblings, children, and grandchildren. They felt that rebuilding close relationships with friends and family members was important. Some advised not to become annoying old people, and others did not want to spend their “seenager” years tied to their grandchildren. Some living in retirement communities suggested minding one’s own business and avoiding quarrels. Tsai [24] analyzed the relationships of senior couples and noted the following: (a) clear division of housework, (b) open communication, (c) clear distinctions between public and private affairs, and (d) high-quality time. These show the importance of rebuilding relationships.Residence in old age: Moving a place of residence in one’s old age is a major event. The interviewees presented polarized views regarding this issue. Younger interviewees wished to live with their spouse, family, and even siblings. Some were not against living in a retirement community if necessary. However, they were also worried that there were only seniors in retirement communities or had to consider their budgets. Some believed that retirement communities were safe and provided many activities. Some even already saw retirement communities as their home. Returnees from overseas compared themselves to migratory birds, with the flexibility to come and go.Significant others: From a psychological perspective, Sullivan [27] defined significant others as people who are extremely crucial to one’s life or wellbeing. These people could be one’s parents, teachers, elders, siblings, friends, peers, or even passersby met by chance. Indeed, many interviewees indicated that they had to consider the opinions of their own elderly parents and their married children into account if they wanted to live in a retirement community. Some shared not only their views or the views of others regarding retirement communities but also their own adaptation experiences or those of other residents.

### 4.2. Interview Implications

Apart from focused interviews with confidential recordings, interviewees glean several significant implications from the content and demeanor of the interviewees’ speech:(1)Baby boomers aspire to age successfully, replacing the traditional notion of “raising children as a hedge against aging”.

In terms of overall discussion, the most highly discussed concern is settling down and interpersonal relationships, with opinions sharply divided. Filial piety for Taiwanese people encompasses expressions of obedience, honoring ancestors, carrying on family lineage, living with and taking care of parents, etc., with living alone often perceived as abandonment by children. However, as times change, attitudes among baby boomers are gradually shifting, sparking lively discussions and a keen interest in understanding the perspectives of other interviewees. Generally, baby boomers entering their later years hope to rebuild close relationships with spouses, relatives, and old classmates, but may not necessarily wish to live with their children, instead focusing more on health, and whether their dreams have been fulfilled.

(2)Residents of health villages share positive stories and new options.

Many residents of health villages share their inner transformations from rejection to acceptance and joy upon living in these communities. Younger women feel that living in health villages is safe and convenient, while older individuals find reassurance in medical care. They share their stories in several focused interviews and spread the word among peers, as user experiences often serve as the best form of publicity. The proliferation of health institutions in Taiwan reflects this trend, although cost remains a significant consideration for residency.

(3)Women demonstrate psychological adaptability.

Pre-retirement men exhibit a high degree of concern regarding the Physical factor, likely because men generally experience more drastic and noticeable changes in physical strength and appearance as they age, leading to greater reflection. In contrast, women show a higher degree of discussion regarding the Prospect and Place and Relationships factors, possibly due to women’s regular focus on health, financial planning, and future arrangements. They tend to be more accepting and adept at handling emotions, playing more caregiving roles within the family, which makes them more adaptable and receptive to approaching old age. Particularly, younger women who have recently retired exhibit enthusiastic expressions when they talked about learning new things.

(4)Being a successful elder is something that requires learning.

Life demands learning at every stage, and it should not be taken for granted. Indeed, a senior long-term care worker expressed that in the face of aging, there is no need to be overly depressed; however, one should not be completely ignorant either. One of the interviewees (82-0703-2-2-2) said, “You have to learn how to be a senior; you won’t know how just by getting old. It’s like being a mother; you won’t know how just by having a baby. That’s impossible; you have to learn how.” This perspective also fully demonstrates their determination to age successfully.

There are several limitations to this study. Firstly, the study was conducted in northern Taiwan, where participants generally have average incomes and education levels and hold relatively open views on the concept of raising children to support old age. Secondly, due to the small sample size, presenting quantitative analysis results may lack persuasiveness. Additionally, participants in focus group interviews may be influenced by the opinions of others and may not fully express their true thoughts. Future follow-up studies could focus on different regions, expand the pool of participants, or conduct more in-depth interviews to explore finer nuances of personal stories.

## 5. Conclusions

This study investigated the perceptions and responses of quasi-seniors with regard to aging. Through focused interviews, the views of 93 interviewees between the ages of 55 and 75 were collected. The four major categories that Phelan established for successful aging (physical, functional, psychological, and social health) were modified to perceptions and response factors suitable for quasi-seniors, namely, the 4P Strategies: Physical, Psychological, Prospect, and Place and Relationships. A brief analysis of this study is as follows: (1) Physical factor: The interviewees were startled at the decline in their stamina and physiological functions, and they had already begun to devise strategies for health preservation and exercise. (2) Psychological factor: Whether or not they could mentally adapt, the interviewees agreed that moderate stress relief was absolutely necessary. (3) Prospect factor: The interviewees felt that one should plan their financials early, contemplate the value of life, and more actively learn and realize one’s dreams. (4) Place and Relationships factor: The interviewees aimed to rebuild their close relationships with their spouses, family members, and old friends and had polarized views regarding where to live in their old age; each also had own opinions and considerations. In analyzing the implications of the interviews, the following was also found: (1) Baby boomers aspire to age successfully, no longer relying on the traditional expectation of “children supporting parents in old age.” (2) Residents of health villages offer positive stories and share new options. (3) Women demonstrate psychological adaptability. (4) Being a successful elder requires learning.

From Phelan’s successful aging to the 4P Strategies of quasi-seniors, this study aimed to help people who are about to enter old age to understand themselves and each other, find the courage to face the many challenges of aging, and have the chance to become happy seniors who thrive with age. It was also hoped that this study could fill in gaps in research involving retired and non-retired individuals between the ages of 55 and 75 and provide a reference for relevant policies.

## Data Availability

The data are available upon reasonable request to the corresponding author.

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
