# Peer review of "Quasi-Seniors’ Perception, Response, and Planning from the Perspective of Successful Aging"

_healthcare, 2024, doi:10.3390/healthcare12070766_

Round 1

Reviewer 1 Report

Comments and Suggestions for Authors

In general, the authors have developed the arguments and the specific analysis from a qualitative perspective. However, from the quantitative analysis, it seems to remain as evidence only the analysis of frequencies of certain terms or expressions, leaving a little in the air, the understanding of how these terms mentioned are inserted in a semantic network constructed in the discourse of the participants. The analysis of these networks could facilitate a deeper understanding not only of the meaning of the variables under analysis, but could also help to elucidate the sense of these for the defined age groups. 

Reviewer 2 Report

Comments and Suggestions for Authors

Good job but there are things to improve:

- in the summary you do not have to put bibliographic citations

- it would be necessary to properly mark the objective in the final part of the introduction

- In the introduction the bibliographic references were missing in each paragraph

- the material and methods section is not well structured

- very long conclusion

To be accepted, a better restructuring of the summary, introduction, material and methods, and conclusion would be necessary.

Encouragement to continue working

Reviewer 3 Report

Comments and Suggestions for Authors

The manuscript reports a mixed(?) method study aimed to "This study adopted Phelan’s four categories, which emphasize involvement and function, in order to understand how  quasi-seniors break age barriers, actively devise coping strategies, and aim for successful aging" (lines 332-4).

The main outcome of the study seems to be a modification of Phelan's model.

I have a major methodological concern about the study, which adopts qualitative and quantitative methods without any justification nor explanation. If the study is a mixed method study, this should be declared and motivated, and a model of mixed method study should be adopted. Overall, I'm very unconfortable with the approach of the present study. If you do qualitative research, you cannot use the findings  as statistical elements. Mixed methods more often alternate a phase of qualitative study, that provide the foundation to develop a valid and reliable instrument (usually a survey or questionnaire) and do a quantitative study. The opposite is possible, as well: start with a questionnaire and then do a qualitative study to deepen the understanding of the numeric results. Other models to mix methods are available, but as far as I know, in no cases you derive numbers from the transcripts of focus groups. Phelan used a survey, that is a tool designed to obtain numerical data.

See: ALMEIDA, Fernando. STRATEGIES TO PERFORM A MIXED METHODS STUDY. European Journal of Education Studies, [S.l.], aug. 2018. ISSN 25011111. Available at: <https://oapub.org/edu/index.php/ejes/article/view/1902/4540>. Date accessed: 10 mar. 2024. doi:http://dx.doi.org/10.46827/ejes.v0i0.1902.

In its present form the article is not acceptable for publication, and I suggest that you provide only the qualitative analysis and rewrite the article according to the SRQR guidelines for reporting. 

There are other issues in the text, I just mention some of them, to help the authors to improve the text:

- the choice of grounded theory should be justified. Actually, it is meant to produce a new theory. Your goal could be to expand or modify Phelan's  model. Was it?

- how were the subjects recruited?

- in the Discussion there is no mention of the cross-cultural aspects of the topic at stake. I suggest that you consider this element in your Discussion 

- there are statements that are not supported by a reference. See for example lines 70-75 or lines 199-200 (does this list come from reference 16?)

Round 2

Reviewer 2 Report

Comments and Suggestions for Authors

Good afternoon.

Good work, I hope it is published soon!!

Keep creating content like this

Reviewer 3 Report

Comments and Suggestions for Authors

I read the revised version of the article and found that the authors solved the issues I mentioned in my first report.

The article is suitable for publication

Comments on the Quality of English Language

The grammar and spelling is correct, some sentences does not "sound" the way they should, but this is a common problem when trying to express complex concepts in another language than own. Overall the meaning is perfectly clear